# Epizoic Rotifers and Microcrustaceans on Bivalves of Different Size and Behavior

**Jolanta Ejsmont-Karabin [1,*], Maciej Karpowicz [2] and Irina Feniova [3]**

[1] Hydrobiological Station in Mikołajki, Nencki Institute of Experimental Biology, Polish Academy of Sciences, 3 Pasteur Street, 02-093 Warsaw, Poland

[2] Department of Hydrobiology, Faculty of Biology, University of Białystok, Ciolkowskiego 1J, 15-245 Bialystok, Poland; m.karpowicz@uwb.edu.pl

[3] Institute of Ecology and Evolution, Russian Academy of Sciences, Leninsky Prospect 33, 119071 Moscow, Russia; feniova@mail.ru

[*] Correspondence: j.karabin@nencki.edu.pl

**Abstract:** Previous mesocosm experiments with the epizoon of *Dreissena polymorpha* revealed that the communities of Rotifera and Crustacea were much more abundant and had higher species richness than epizoon of *Unio tumidus* in nature. These differences could be attributed to different environmental conditions and/or different host behavior. To test this hypothesis, we compared epizoon of *D. polymorpha* and *U. tumidus* placed in identical mesocosm conditions, in which *Unio* could not move vertically due to the lack of sediments. Half of the mesocosms contained *D. polymorpha*, the other half comprised *U. tumidus*. Each species of mollusks was kept in the mesocosms with eutrophic and mesotrophic conditions. Finally, we established four treatments that were replicated in triplicate mesocosms. Mesocosm experiments showed that epizoon communities of *U. tumidus* were even more abundant than that of *D. polymorpha* and their species richness was similar. Therefore, we concluded that previously revealed strong differences between epizoon communities of *D. polymorpha* and *U. tumidus* were related to the different environmental conditions and bivalve behavior.

**Keywords:** epizoon; Rotifera; Crustacea; *Dreissena polymorpha*; *Unio tumidus*

## 1. Introduction

Mollusks serve as an excellent substratum for epizoic rotifers and microcrustaceans because epizoon is provided by additional habitat, takes advantage of the host ability to aggregate food, and is better protected against grazing [1]. There are many reports on the colonization of animals by sessile and free-swimming rotifers [2,3]. However, data on rotifer epizoon of mollusks are rare. The reports on epizoic microcrustaceans in freshwaters are also very few [4].

Ejsmont-Karabin and Karpowicz [3] revealed surprisingly high densities of epizoic rotifers on *Dreissena polymorpha*. They showed that large cladocerans or small fish may force plankton and littoral rotifers to hide in the druses of *D. polymorpha*. However, in the treatment with the presence of large cladocerans and fish the density of epizoic rotifers was lower relative to treatments without cladocerans and fish. Ejsmont-Karabin and Karpowicz [3] suggested that such refuges which are easily accessible to cladocerans or small fish may become a trap for rotifers. Similarly, macrophytes are regarded as risky areas for zooplankton due to presence of littoral predators [5,6]. A preliminary report on the secondary colonization of zebra mussels by crustacean zooplankton also suggested that epizoon community could be rich in species and poor in abundance [7].

Ejsmont-Karabin and Karpowicz [3] provided evidence to show that epizoon communities of Rotifera on *Dreissena polymorpha* were more abundant and consisted of more species than epizoon on *Unio tumidus* [8]. We proposed that these differences could be related to the different environmental conditions. In particular, in the case of *D. polymorpha*,

the mollusks were kept in the mesocosms, while *U. tumidus* mussels were taken directly from the natural environment. It is noteworthy that the movement behavior of these two species of mussels is different. Specifically, unionids, in contrast to zebra mussels, move horizontally and burrow in sediments that can damage rotifers [9].

The goal of the current study was to compare epizoon communities on *Dreissena polymorpha* and *Unio tumidus* placed in identical mesocosm conditions, in which *U. tumidus* could not move vertically due to the lack of sediment in the mesocosms. We believed that in identical environmental conditions and similar behavior of hosts, their epizoon abundance and structure would be similar.

## 2. Materials and Methods

The mesocosm experiments were carried out from 24 June to 1 August 2018. *Dreissena polymorpha* and *Unio tumidus* were collected from the nearby Lake Boczne (the Great Masurian Lakes, north-eastern Poland). They were gently brushed under tap water to remove the epizoon and then placed in the appropriate mesocosms filled with 270 L unfiltered lake water. The sediment in the mesocosms were absent.

Samples of ten individuals of *Dreissena polymorpha* and one individual of *Unio tumidus* (and, concurrently, 1 L zooplankton samples) were taken from the mesocosm on Day 1, Day 10, Day 20, Day 30, and Day 40. Epizoic rotifers and microcrustaceans were removed from the shells of the bivalves with a soft bristle brush, transferred into bottles and fixed in a 2% formaldehyde solution. The collected material was analyzed under the light microscope to identify and count epizoon species. Rotifers were identified to the species using appropriate identification keys [10–17] and Rotifer World Catalog [18]. Identification of crustacean species was based on Janetzky et al. [19], Flößner [20], Błędzki and Rybak [21] etc.

Bivalve surface was measured, and the density of epizoon was expressed in number of ind. 100 cm$^{-2}$ on the shell surface.

We established four treatments that were replicated in triplicate mesocosms:

UtM treatment (three mesocosms)—15 individuals of *U. tumidus* per mesocosm (about 24 ind. m$^{-2}$), water taken from mesotrophic Lake Majcz Wielki;

UtE treatment (three mesocosms)—15 individuals of *U. tumidus* per mesocosm (about 24 ind. m$^{-2}$), water taken from eutrophic Lake Jorzec;

DpM treatment (three mesocosms)—150 individuals of *D. polymorpha,* (about 240 ind. m$^{2}$), water taken from mesotrophic Lake Majcz Wielki;

DpE treatment (three mesocosms)—150 individuals of *D. polymorpha,* (about 240 ind. m$^{2}$), water taken from eutrophic Lake Jorzec.

Lake Majcz Wielki is a dimictic mesotrophic lake with surface area of 164 ha and maximum depth of 16.4 m [22]. Lake Jorzec is a eutrophic stratified lake with surface area of 42 ha and maximum depth of 11.6 m [23]. Both lakes are situated in the Jorka River watershed, north-eastern Poland.

We did a cursory review of samples from both lakes to determine species composition of zooplankton on the day of water sampling. Rotifera communities in Lake Majcz Wielki were dominated by pelagic species *Keratella cochlearis* and *Polyarthra remata*. The same species occurred in zooplankton of Lake Jorzec, but there were differences in abundance of *Keratella cochlearis*. Two other abundant species in Lake Jorzec were *Polyarthra vulgaris* and *Pompholyx sulcata*. Crustacean zooplankton in both lakes was dominated by *Ceriodaphnia pulchella*, *Diaphanosoma brachyurum*, *Bosmina longirostris*, *Thermocyclops oithonoides* and *Mesocyclops leuckarti*.

The differences in zooplankton community parameters between environmental conditions (host, trophic, day of the experiment) were tested using a two-way analysis of variance (ANOVA) with Type III SS (sum of squares). The statistical analyses were performed with XLSTAT2020 (Addinsoft) and Biodiversity Pro: Free Statistics Software for Ecology (Software Informer). A probability level of $p < 0.05$ was considered significant.

## 3. Results

The density of the epizoic rotifers on the mussels in the mesocosms was much higher than that on the mussels collected from Lake Boczne. Whereas the average density of Monogononta rotifers in the epizoon of *U. tumidus* in the lake was 39–54 ind. 100 cm$^{-2}$, their density in the experimental mesocosms constituted ca. 2200 ind. 100 cm$^{-2}$. The average densities of the epizoon of *D. polymorpha* were ca 158 ind. 100 cm$^{-2}$ and ca. 1520 ind. 100 cm$^{-2}$ in the lake and mesocosms, respectively. Similarly, the densities of the microcrustaceans on the mussels were much higher in the mesocosms than in the lake. The maximum density of crustaceans on *U. tumidus* was 3.2 ind. 100 cm$^{-2}$ and 63.8 ind. 100 cm$^{-2}$ in the lake and mesocosms, respectively. The maximum density of crustaceans on *D. polymorpha* was also higher in the mesocosms (225.9 ind. 100 cm$^{-2}$) than in the lake (14.7 ind. 100 cm$^{-2}$).

The maximum density of Bdelloida was recorded on Day 10 and they were more abundant in the epizoon of *D. polymorpha* than on *U. tumidus* (Figure 1a). The maximum density of Monogononta epizoon on *D. polymorpha* was recorded on Day 20, and it was similar or higher than that in the epizoon on *U. tumidus* (Figure 1b). The densities of microcrustaceans began to increase on Day 20 and were still continuing to rise on termination of the experiment on Day 40 (Figure 1c). There were no significant differences in the densities of microcrustacean epizoon either between mesotrophic and eutrophic conditions (F = 0.025; $p$ = 0.89) or between host mollusk species (F = 4.0; $p$ = 0.07). However, at the end of the experiment, microcrustaceans were more abundant on *D. polymorpha* than on *U. tumidus* (Figure 1c).

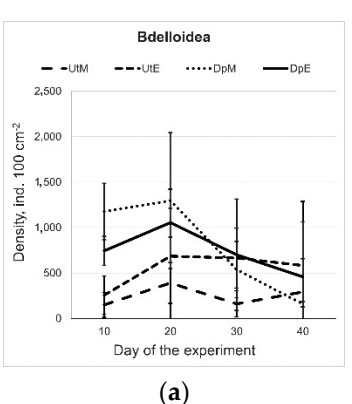
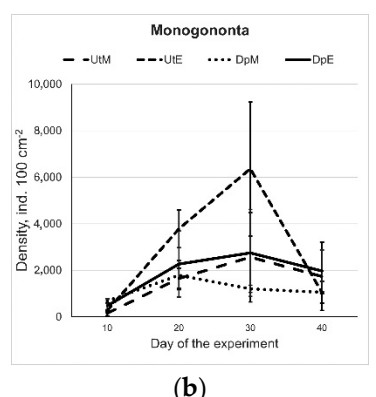
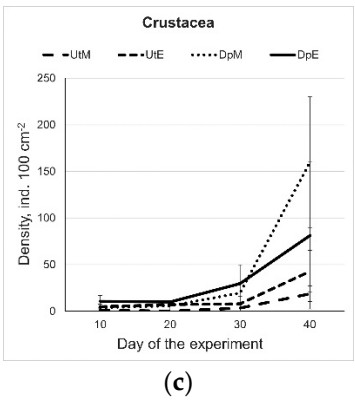

(a)                                     (b)                                     (c)

**Figure 1.** Density dynamics (ind. 100 cm$^{-2}$) of epizoic rotifers, i.e., Bdelloidea (**a**) and Monogononta (**b**), and microcrustaceans (**c**) on *D. polymorpha* and *U. tumidus* in the mesotrophic and eutrophic experimental conditions (Experimental treatment abbreviations: UtM—*Unio tumidus* with mesotrophic water source, UtE—*U. tumidus* with eutrophic water source, DpM—*Dreissena polymorpha* with mesotrophic source, DpE—*D. polymorpha* with eutrophic water source.

In the experiment, 85 species of Rotifera were identified (See details in supplementary). Species richness on *D. polymorpha* and *U. tumidus* was rather similar, specifically, 71 and 77, respectively. Exclusive species for *D. polymorpha* (8) or *U. tumidus* (15) were occurring sporadically at very low densities. Species compositions were similar on both species of mollusks. The most abundant species were representatives of genera *Lecane* (*L. closterocerca, L. lunaris, L. luna* and *L. flexilis*), *Lepadella* (*L. quadricarinata*), *Colurella* (*C. colurus* and *C. adriatica*) and *Trichocerca* (*T. porcellus*). There were no significant differences in species richness of epizoic rotifers between the lakes of different trophic status. Seventy-eight species were recorded in the eutrophic lake and sixty-nine species were found in the mesotrophic lake.

Rotifer abundance was gradually growing in all the treatments. After they achieved maximum, their abundance decreased (Table 1; Figure 2). This trend was observed for most of species except *L. luna* and *L. lunaris* in the mesotrophic conditions, which increased until

the end of the experiment. In eutrophic conditions only *L. lunaris* and *L. luna* from epizoon of *D. polymorpha* increased their density until the end of the experiment. Total and mean species richness of Rotifera first gradually increased and then decreased until the end of the experiment in all four treatments (Table 1). Most rotifer species achieved markedly higher maximum densities in eutrophic conditions relative to mesotrophic mesocosms (Figure 2) except *Lecane flexilis* which was more abundant in mesotrophic conditions.

**Table 1.** Total and mean (±standard deviation) species richness of rotifers and microcrustaceans in epizoon on *Dreissena polymorpha* and *Unio tumidus* in the experimental treatments. Abbreviations: UtM—*U. tumidus* with mesotrophic water source, DpM—*D. polymorpha* with mesotrophic water source, UtE—*U. tumidus* with eutrophic water source, DpE—*D. polymorpha* with eutrophic water source.

| | | Rotifer Species Richness | | | | Microcrustacean Species Richness | | | |
|---|---|---|---|---|---|---|---|---|---|
| | | Day 10 | Day 20 | Day 30 | Day 40 | Day 10 | Day 20 | Day 30 | Day 40 |
| UtM | Total | 19 | 43 | 36 | 25 | 1 | 0 | 3 | 5 |
| | Mean | 10 ± 5 | 25 ± 8 | 25 ± 4 | 14 ± 1 | 0.3 ± 0.6 | 0 | 1.0 ± 1.0 | 2.3 ± 0.6 |
| UtE | Total | 24 | 55 | 44 | 26 | 4 | 4 | 5 | 6 |
| | Mean | 13 ± 3 | 33 ± 7 | 27 ± 2 | 15 ± 4 | 1.3 ± 0.6 | 1.3 ± 0.6 | 1.7 ± 1.5 | 2.7 ± 0.6 |
| DpM | Total | 25 | 42 | 41 | 35 | 2 | 2 | 4 | 7 |
| | Mean | 18 ± 6 | 27 ± 4 | 28 ± 2 | 22 ± 4 | 1.0 ± 1.0 | 1.0 ± 1.0 | 2.0 ± 1.0 | 3.7 ± 0.6 |
| DpE | Total | 25 | 44 | 43 | 37 | 5 | 5 | 7 | 9 |
| | Mean | 14 ± 3 | 27 ± 10 | 26 ± 9 | 18 ± 4 | 1.3 ± 0.6 | 2.0 ± 0.7 | 3.7 ± 1.5 | 3.7 ± 0.6 |

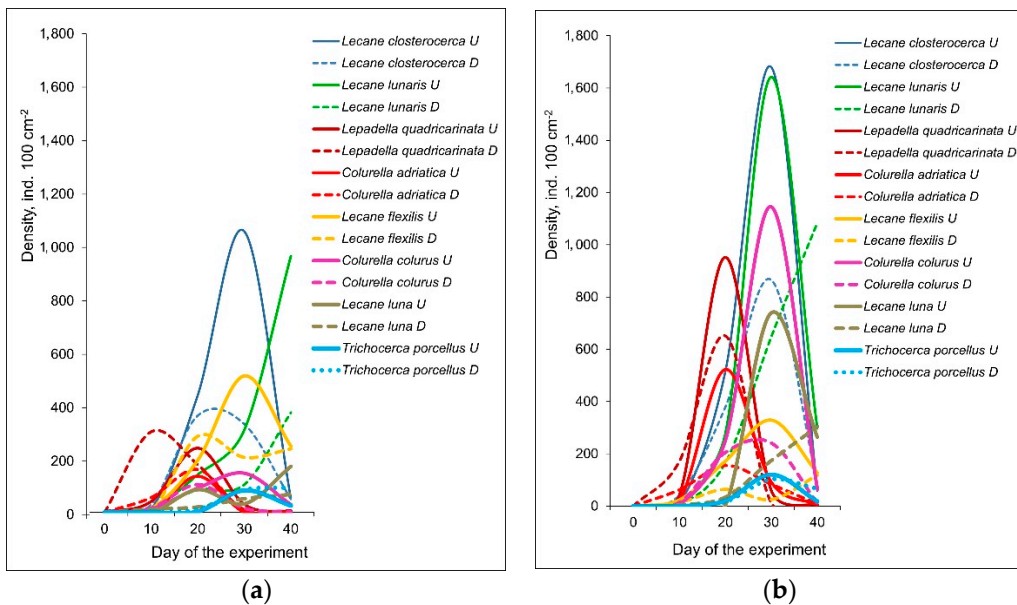

**Figure 2.** Abundance dynamics (ind. 100 cm$^{-2}$) of the most abundant rotifer species in the mesotrophic (**a**) and eutrophic (**b**) experimental conditions. Abbreviations after species names: U—*Unio tumidus*, D—*Dreissena polymorpha*.

Species composition of rotifers was similar in the epizoon communities on *D. polymorpha* and *U. tumidus*. However, the abundance of rotifers was significantly higher on *U. tumidus* relative to that on *D. polymorpha* (Figure 2), except *Trichocerca porcellus,* which was more abundant on *D. polymorpha*. Three rotifer species characteristics for epizoon of bivalves (*Lecane pumila*, *Lophocharis naias* and *Wulfertia* sp.) [24] were present in all four treatments (Table 2).

**Table 2.** Maximum density (ind. 100 cm$^{-2}$) of epizoic rotifers on bivalve mollusks.

|  |  | Day 10 | Day 20 | Day 30 | Day 40 |
|---|---|---|---|---|---|
| UtM | *Lecane pumila* | - | 20 | 57 | - |
|  | *Lophocharis naias* | 3 | - | - | - |
|  | *Wulfertia* sp. | - | - | 140 | - |
| UtE | *Lecane pumila* | 64 | 28 | 130 | - |
|  | *Lophocharis naias* | - | 7 | - | - |
| DpM | *Lecane pumila* | 28 | 4 | - | 4 |
|  | *Lophocharis naias* | 165 | 25 | 15 | 31 |
| DpE | *Lecane pumila* | - | - | 9 | 6 |
|  | *Lophocharis naias* | 117 | 61 | 14 | 6 |
|  | *Wulfertia* sp. | - | - | 4 | - |

In total, we identified 20 species of Crustacea (15 Cladocera, 2 Cyclopoida, 1 Calanoida, 2 Harpacticoida) (See details in supplementary). Species richness of crustacean species differed significantly between the host mollusks (F = 19.9; $p$ = 0.001) and between trophic conditions (F = 13.6; $p$ = 0.004). The species richness of crustaceans was higher on *D. polymorpha* than on *U. tumidus* and this index was greater in eutrophic conditions than in mesotrophic mesocosms (Table 1). However, a similar set of crustacean species was found on both hosts. *Chydorus sphaericus* was the most frequent species and its abundance increased during the experiment (Figure 3a), even to 154 ind. 100 cm$^{-2}$ (Figure 3). The abundance of *C. sphaericus* did not differ significantly between the host mollusks and between as well as different trophic conditions. The other microcrustacean species characteristic for mussels epizoon in our experiments were *Nitokra hibernica*, *Acroperus harpae*, *Coronatella rectangula*, *Pleuroxus aduncus*, *Pleuroxus leavis*, and *Camptocercus rectirostris* (Figure 3b). In comparison, we found only 4 crustacean species on the mussels in the lake, two of which (*Nitokra psammophila*, and *Monospilus dispar*) were not available in the epizoon in the mesocosms. In the mesocosms, there were crustaceans in the epizoon communities which were also abundant in the plankton (*Ceriodaphnia pulchella*, *Diaphanosoma brachyurum*, *Mesocyclops leuckarti*, and *Thermocyclops oithonoides*). Some crustacean species (*Alona affinis*, *Alona guttata*, *Alonella nana*, *Bosmina longirostris*, *Polyphemus pediculus*, *Scapholeberis mucronata*, and *Eudiaptomus graciloides*) were rare in the epizoon.

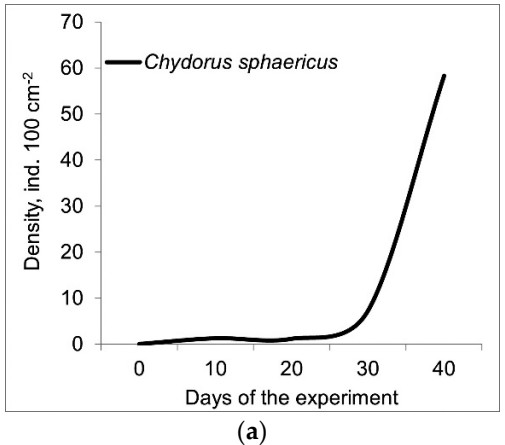
(a)

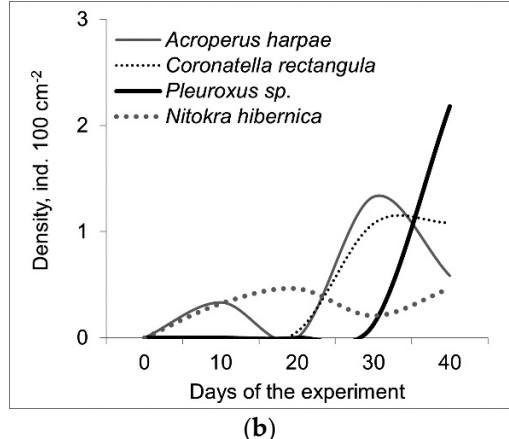
(b)

**Figure 3.** Abundance dynamics (ind. 100 cm$^{-2}$) of *Chydorus sphaericus* (**a**) and other most abundant crustacean species (**b**) (averaged over the four treatments).

## 4. Discussion

We showed that under identical experimental conditions, epizoon communities of *U. tumidus* were more abundant than those of *D. polymorpha* and their species structure were similar. These results have supported the hypothesis which stated that epizoon communities of *D. polymorpha* and *U. tumidus* are dependent on the environmental conditions [3].

However, Ejsmont-Karabin and Karpowicz [3] have noticed before that epizoic rotifer communities on *D. polymorpha* in the experimental conditions were more abundant and their species richness was higher than on *U. tumidus* in nature [8]. They suggested that this difference could be related to different behavior of these two mussel species. Unionids, in particular, commonly move horizontally and burrow in sediments; thus, they may damage rotifers [9]. However, in nature, adult individuals of *Unio tumidus* were mainly found at the sediment surface because they do not burrow so frequently in contrast to juveniles [25]. Therefore, epizoon communities could grow on the shells of *U. tumidus*. There may be also another reason affecting epizoon in nature. Ejsmont-Karabin and Karpowicz [3] suggested that low species richness of rotifers in epizoon in nature could be attributed to predator pressure which is often high in benthic food webs [26]. High predatory pressure of macroinvertebrates on crustaceans [5] can be attributed to high accessibility of prey accumulating on bivalve shells and especially in *Dreissena*'s druses [27].

Abundances of different taxa in the experiment reached peaks in accordance with the hatching process described by Kalinowska et al. [28]. In their experiment, species forming resting cysts such as bdelloid rotifers hatched after the first day of their incubation whereas hatching from resting eggs (monogonont rotifers and crustaceans) occurred from the 2nd to 3rd day. Similarly to our experiment, abundance of the three groups of invertebrates first gradually increased and then sharply decreased. Rotifer species reached maximum abundance on the 23rd day, cladocerans attained peak on the 21st day and copepods—on the 33rd day [28].

The decrease of the rotifers in the experiment could be associated with the growth of microcrustaceans feeding on the mussel periphyton and thus mechanically destroying rotifers. At the beginning of the experiment, the epizoon was mainly represented by planktonic crustaceans while in the second half of the experiment, Chydoridae including *Chydorus sphaericus*, *Alona* spp., *Pleuroxus* spp., *Acroperus harpae*, and *Camptocercus rectirostris* dominated. Chydoridae are poor swimmers and commonly feed creeping along submerged surfaces (macrophytes or bottom substrates) [29]. The significant increase of periphyton grazers at the end of the experiment may be also the cause of sessile rotifers absence.

*Chydorus sphaericus* was the most abundant in the epizoon at the end of the experiment while it was rare in the plankton. In contrast to other chydorid species, *Ch. sphaericus* exerts two alternative behavioral traits. Specifically, it inhabits littoral zones with macrophytes and bottom substrates [29,30] and also commonly occurs in the plankton of highly eutrophic lakes with Cyanobacteria blooms [31,32]. We additionally showed that *C. sphaericus* can be highly abundant in the epizoon of the mussels at the later stages of its development.

The decrease of the rotifers in the experiment might be related to the diet of both mussels. Makhutova et al. [33] showed that *U. tumidus* and *Dreissena* species obtained food of different qualities. In their experiments *Dreissena* consumed plankton species, i.e., more-valuable food, while *U. tumidus* fed on detritus and phytobenthic species. Although the diet of both species consisted mainly of algae and detritus enriched with bacteria [34,35], zooplankton can also be a potential food source. Rotifers had suitable size range, but they would not have been able to reach such high numbers if they had been an essential part of their hosts' diet.

The results of the secondary colonization of the mussels in the current experiment revealed that zooplankton communities developed in identical environment were similar. This conclusion is in accordance with the data on aquatic plants in the littoral zone of the lakes, which showed that zooplankton community structure developed in identical macrophyte habitat was similar [36,37].

High species richness recorded in the epizoon communities of *D. polymorpha* and *U. tumidus* in the experiment have supported hypothesis of functional redundancy, which states that some species play similar roles in communities and may therefore be substitutable with little impact on ecosystem processes [38]. At the beginning of the experiment, some species existed at the stages of resting or subitaneous eggs hidden in the recesses of shells. This explains the great variety of species of rotifer assemblages in nature. Abundant

species of the epizoon in the experiment were similar in terms of body size, behavior and food requirements, i.e., they occupied similar niches. Their coexistence is facilitated by spatial and temporal environmental variability which may provide room for functional redundancy at small spatial and temporal scales [39]. Frequent disturbances caused by physical or biotic instability in nature facilitate coexistence of species with similar ecological niches [40].

Freshwater mussels are extremely important for supporting life of different invertebrate species [41]. Aldridge et al. [42] reported that the diversity of associated macroinvertebrates is markedly higher at sites with higher densities of mussels. Results of our experiment show that the mussels play also a very important role as hot spots of rotifer and crustacean diversity.

**Supplementary Materials:** The following supporting information can be downloaded at: https://www.mdpi.com/article/10.3390/d14040293/s1. A list of Rotifera and Crustacea from epizoon on Dreissena polymorpha and Unio tumidus.

**Author Contributions:** Conceptualization, J.E.-K. and M.K.; methodology, J.E.-K., M.K. and I.F.; investigation and writing—original draft preparation, J.E.-K., M.K.; writing—review and editing, J.E.-K. and I.F. All authors have read and agreed to the published version of the manuscript.

**Funding:** This research was supported by the Polish National Science Centre by grant number 2016/21/B/NZ8/00434.

**Institutional Review Board Statement:** Not applicable.

**Informed Consent Statement:** Not applicable.

**Data Availability Statement:** Not applicable.

**Conflicts of Interest:** The authors declare no conflict of interest.

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
