# Peer review of "Epizoic Rotifers and Microcrustaceans on Bivalves of Different Size and Behavior"

_diversity, doi:10.3390/d14040293_

Round 1
Reviewer 1 Report
Colonization by rotifers and crustaceans is due to many factors including escape from active predators, energy-saving, camouflage with surface of hosts and other biological features (such as tube-building). The authors in this work have addressed mostly escape from predation or avoidance from competitors. Thus, the introduction requires some more expansion of ideas as the predation or competition alone cannot explain the epizoic preferences of certain zooplankton species. It is also important that the nature of surface of the molluscan shells used in this study may be considered for discussion.
Most species settled on the surface of molluscs are littoral-benthic which may simply settle down on possibly any substrate (stone, dead plants, dead molluscan shells, etc.). To prove that crustacean and rotifer settlement on molluscs is to avoid predation, experimental design should be different (introducing predators such as insect larvae, fish larvae and/or predatory copepods).
In the test design, I am unable to see a control (medium without Dreissena or Unio). If the surface area of molluscs has any role is to be tested statistically. For example, the size of the shells used in the beginning may be different (smaller) than that later (after 40 days). Possibly increase in rotifer density (and taxa) may be higher with increase in surface area.
The authors need to provide a list of zooplankton species encountered in mesocosm (as an annex).
The last sentence of the manuscript (227 – 229) requires a citation as the study did not contain data in relation to this statement.
Author Response
Reviewer 1:
Remark: Colonization by rotifers and crustaceans is due to many factors including escape from active predators, energy-saving, camouflage with surface of hosts and other biological features (such as tube-building). The authors in this work have addressed mostly escape from predation or avoidance from competitors. Thus, the introduction requires some more expansion of ideas as the predation or competition alone cannot explain the epizoic preferences of certain zooplankton species. It is also important that the nature of surface of the molluscan shells used in this study may be considered for discussion.
Response: The assumptions of this experiment mean that predation and behavior can only be treated as a kind of "black box". The work is also not about the influence of different predators on the bivalve epizoon, but about the fact that, deprived of predator pressure and placed in the same conditions, different species of bivalve mollusks are inhabited by similar epizoon communities in terms of numbers and species structure.
Remark: Most species settled on the surface of mollusks are littoral-benthic which may simply settle down on possibly any substrate (stone, dead plants, dead molluscan shells, etc.). To prove that crustacean and rotifer settlement on mollusks is to avoid predation, experimental design should be different (introducing predators such as insect larvae, fish larvae and/or predatory copepods).
Response: This is actually not what we wanted to show. Our goal was to show that epizoic rotifers and crustaceans settle for some reason (e.g. food concentrated due to filtering activity of mussels), and that in nature they are under strong predator pressure which decrease their numbers . We do not claim in our work that the epizoic way of life is an escape from predators.
Remark: In the test design, I am unable to see a control (medium without Dreissena or Unio).
Response: The treatment without mollusks is not necessary because we compared epizoon between two bivalve species.
Remark: If the surface area of molluscs has any role is to be tested statistically. For example, the size of the shells used in the beginning may be different (smaller) than that later (after 40 days). Possibly increase in rotifer density (and taxa) may be higher with increase in surface area.
Response: The individuals were selected randomly, and size of hosts did not influence rotifers and crustaceans densities, because bivalves were measured, and then the density of epizoon was expressed in units of the shell surface. This information we added into Methods.
Remark: The authors need to provide a list of zooplankton species encountered in mesocosm (as an annex).
Response: Yes, you are right. It was added as recommended.
Remark: The last sentence of the manuscript (227 – 229) requires a citation as the study did not contain data in relation to this statement.
Response: OK. Done.
Reviewer 2 Report
This is nice paper and deserves publication.
Given to the fact that your survey clearly confirms the important role of freshwater mussels in providing suitable habitat for numerous macro invertebrate species, including rotifers and copepods it would be nice if you referring to some publications (e.g.. Sousa R, Halabowski D, Labecka AM, et al. The role of anthropogenic habitats in freshwater mussel conservation. Glob Change Biol. 2021;27:2298–2314. https://doi.org/10.1111/gcb.15549 or Lopes-Lima M, Sousa R, Geist J, et al. 2017. Conservation status of freshwater mussels in Europe: state of the art and future challenges. Biol Rev 92: 572–607) after the line 50 and before last paragraph of Introduction, address the conservation needs for the freshwater mussels as extremely important for supporting the other invertebrate life.
Author Response
Reviewer 2
Remark: Given to the fact that your survey clearly confirms the important role of freshwater mussels in providing suitable habitat for numerous macroinvertebrate species, including rotifers and copepods it would be nice if you referring to some publications (e.g.. Sousa R, Halabowski D, Labecka AM, et al. The role of anthropogenic habitats in freshwater mussel conservation. Glob Change Biol. 2021;27:2298–2314.
Response: We would like to refer to this work. Unfortunately, this publication does not deal with the role of mussels in providing a suitable habitat for epizoic organisms, but is related to the role of artificial reservoirs in providing a habitat for molluscs.
Remark: or Lopes-Lima M, Sousa R, Geist J, et al. 2017. Conservation status of freshwater mussels in Europe: state of the art and future challenges. Biol Rev 92: 572–607) after the line 50 and before last paragraph of Introduction, address the conservation needs for the freshwater mussels as extremely important for supporting the other invertebrate life.
Response: …but we decided to add this piece of text at the end of the manuscript: “Freshwater mussels are extremely important for supporting life of different invertebrate species [26]. Aldridge et al. [27] reported that the diversity of associated macroinvertebrates is markedly higher at sites with higher densities of mussels. Results of our experiment show that the mussels play also a very important role as hot spots of rotifer and crustacean diversity.”
Reviewer 3 Report
I have provided a reviewer report for points which will need to be addressed.

Author Response
Reviewer 3
Reviewer’s Summary: This manuscript is well-written and provides new data to clarify why differences in epizoic rotifers and microcrustaceans appear in zebra mussels and benthic freshwater mussels in Polish lakes. The authors showed that when sediment and hence burrowing were removed under experimental conditions, rotifer accumulation on freshwater mussels increased. The experiment is rather simple and was effective in testing hypotheses presented. The manuscript could however, benefit from some editorial, topical, and structural refinement to improve its readability and scientific accuracy.
Response: Thank you for your enormous effort to improve the manuscript.
Suggestions for improvement:
Abstract
Remarks for Line 12, 20: Done. Thank you for the remarks.
Remark: Lines 27-29: Please justify your statement by explaining in a bit more detail how epizoon is “better protected” against grazing and support this statement with published references. I am not sure I agree that epizoic rotifers and microcrustaceans are “better protected” against grazing when on molluscs.
Response: We don't believe it either. This was the opinion presented in the cited publication (JEK: I forgot to include this citation in the text). Our work seems to indicate that this opinion may be false.
Remark: Also, would bivalves not be feeding on rotifers and microcrustaceans when they get near siphons? Please address this in the introduction and as a point for discussion.
Response: JEK: Probably not. Considering the efficiency of water filtration by bivalves, if it were so, in the littoral there would not be any rotifers. On the other hand, the epizoon community sticks to the thin layer of water surrounding the shells.
Remarks to Line 36, 37, 43, 44, 48, 49, 51: OK. Done
Remarks: Line 45 (and throughout remainder of manuscript): provide the species name of Unio and Line 51 (and throughout the remainder of the manuscript): add the species name of Dreissena
Response: OK. Done.
Materials and Method
Remark: Line 58: after “2018”, add “in Poland”
Response: We decided to give this information in the next line after “lake Boczne”.
Remark: Lines 59-60: You describe gently washing mussels with tap water to remove epizoon. Yet, when you wanted to remove epizoon after mussels had been in mesocosms, they were brushed. I have a few issues with this. First, why did you use two different methods to remove epizoon, or did you? Second, would epizoon be damaged by brushing? You suggest rotifers are potentially damaged from burrowing in sediment, yet you abrasively brush the shells to remove them. Would rotifers damaged from brushing create difficulties in identifying them? Please clarify in the methods and add as a discussion point.
Response: Of course the mussels were treated in the same way. We do not know, why we described the procedures as different ones. Thus, we have changed this part of the text.
Remark: Lines 67-68: You mention that “sediments were analyzed under light microscope”, yet in the description of your experimental design, you state that there was no sediment in the mesocosms. Please clarify.
Response: Our mistake. To clarify that we changed this sentence and this part of the text is now – “Epizoic rotifers and microcrustaceans were removed from the shells of the bivalves with a soft bristle brush, transferred into bottles and fixed in a 2% formaldehyde solution. The collected material was analyzed under the light microscope to identify and count epizoon species. Bivalve surface was measured, and the density of epizoon was expressed in number of ind. 100 cm-2 on the shell surface.”
Remark: Lines 82-87: provide published references for the information on rotifer and crustacea zooplankton communities present in these lakes
Response: We added the sentence “We did a cursory review of samples from both lakes to determine species composition of zooplankton on the day of water sampling.” This explains that information on zooplankton communities in these lakes was achieved directly from our research, not from literature.
Remark: Lines 88-92: Provide significance level (e.g., P
Response: These were single samples.
Results
Remarks: Line 94, 104, 105, 127: OK. Done
Remark: Lines 107-108: You can re-write the sentence for sake of accuracy with the presented data to “The densities of microcrustaceans began to increase on Day 20 and were still continuing to rise on termination of the experiment on Day 40.
Response: Done. Now it is much better. Thank you.
Remark: Lines 114-116: Figure 1: in the figure caption, provide definitions of the abbreviations appearing in the graphs (UtM, UtE, DpM, DpE). You could add a period (.) after “conditions” and write “Experimental treatment abbreviations: UtM – Unio tumidus with mesotrophic water source, UtE – U. tumidus with eutrophic water source, DpM – Dreissena polymorpha with mesotrophic source, DpE – D. polymorpha with eutrophic water source.”
Response: Done.
Remark: Figure 2: I don’t understand how you combined abundance dynamics of rotifers for the host species of mussel. I understand the differences between trophic status, but did you average abundances of the various rotifer species for the two host mussel species or add abundances together? It would be better if you could provide side by side comparisons of the four treatments and place the various rotifer species abundances in separate graphs. You could have graphs of the various rotifer species stacked as columns with one column for each treatment. The scheme would be a 4 x 8 series of graphs with the four experimental treatments each as a separate column with 8 species of rotifer abundance as rows. You can provide the four treatments as abbreviations and define as my suggestion for Figure 1. • Figure 3: I don’t understand this figure either. I think you can delete it and follow my suggested scheme for Figure 2, drawn out below. • Figure 4: follow the same scheme I suggested for Figure 2 for this figure. This would then become Figure 3. The scheme would be a 4 x 5 series of graphs, shown below. New Figures 2 and 3 would look something like this: UtM DpM UtE DpE Figure 2. Abundance (ind. 100 cm-2 ) of rotifer species on Unio tumidus and Dreissena polymorpha in mesocosms with water sourced from lakes with varying trophic conditions. Each column represents an experimental treatment, with abbreviations: UtM – U. tumidus with mesotrphic water source, DpM – D. polymorpha with mosotrophic water source, UtE – U. tumidus with eutrophic water source, DpE – D. polymorpha with eutrophic water source. Species of rotifer: a-d) Lecane closterocerca, e-h) Lucane Lunaris, i-l) Lepadella quadricarinata, m-p) Colurella adriatica, q-t) Lecane flexilis, u-x) Colurella colurus, y-bb) Lecane luna, cc-ff) Trichocerca porcellus. The y-axis in each graph is Density (ind. 100 cm-2 ). The x-axis in each graph is Day of the experiment. a) b) c) d) e) f) g) h) i) j) k) l) m) n) o) p) q) r) s) t) u) v) w) x) y) z) aa) bb) cc) dd) ee) ff) UtM DpM UtE DpE Figure 3. Abundance (ind. 100 cm-2 ) of microcrustacean species on Unio tumidus and Dreissena polymorpha in mesocosms with water sourced from lakes with varying trophic conditions. Each column represents an experimental treatment, with abbreviations: UtM – U. tumidus with mesotrphic water source, DpM – D. polymorpha with mosotrophic water source, UtE – U. tumidus with eutrophic water source, DpE – D. polymorpha with eutrophic water source. Species of microcrustacean: a-d) Acroperus harpae, e-h) Coronatalla rectangula, i-l) Chydorus sphaericus, m-p) Pleuroxus sp., q-t) Nitokra hibernica. The y-axis in each graph is Density (ind. 100 cm-2 ). The x-axis in each graph is Day of the experiment. •
Response: No, it wasn't a good idea.The figures became too complicated.Most importantly, however, it was not possible to compare the abundance of species depending on the host species or trophic state.Due to the low densities of Crustacea, separating species also did not make sense.I am asking you for the acceptance of the figures after the changes I made (JEK)
Remark: Line 120 and Lines 152-153: It would be good for you to provide a full list of species of rotifer and microcrustacean identified in a supplementary file or additional table
Response: Yes, you are right. We have done it.
Remark: Lines 122-123: The statement “Species compositions were identical on both species of mollusks” contradicts the previous sentence (Lines 121-122) where you indicate there were species of Rotifera which were exclusive to D. polymorpha and U. tumidus. Please clarify.
Response: We changed “identical” to “similar”.
Remark: Table 1: In the title of the table, please write what the ± values refer to. I presume they are standard error values, but if standard deviation, please clarify.
Revise the table title to “Table 1. Total and mean (± standard error) species richness of rotifers and microcrustaceans in epizoon on Dreissena polymorpha and Unio tumidus in the experimental treatments. Abbreviations: UtM – U. tumidus with mesotrphic water source, DpM – D. polymorpha with mosotrophic water source, UtE – U. tumidus with eutrophic water source, DpE – D. polymorpha with eutrophic water source.” •
Response: Done.
Remarks: Line 142, 157, 160, 187, 208, 209, 216: OK. Done.
Remark: Table 2: provide dashes in empty cells
Response: Done.
Discussion
Remark: Lines 198-199: is there evidence of microcrustaceans feeding on rotifers to back this statement up or are you speculating. If there is evidence, please provide published references in support.
Response: It is some kind of speculation. However, what is more important, this part of the text says “microcrustaceans feeding on the mussel periphyton and thus mechanically destroying rotifers”. Thus it is not on microcrustaceans feeding on rotifers, but on microcrustaceans destroying them mechanically.
Remark: discuss habitat partitioning in lakes in relation to where the various species of rotifers and host bivalves are found;
I think there may be differences at varying depths so that deeper dwelling freshwater mussels might not come into contact with some rotifers or microcrustaceans, or was this not an issue because the freshwater mussels and zebra mussels were collected from the same depths?
Response: We chose Lake Boczne just because both species of bivalves were abundant there at the same (shallow) depth
Remark:What can you say about the possibility that benthic predators may be feeding on epizoon on Unio tumidus compared to Dreissena polymorpha? I would like to know whether there are known predators in the sediments that could be another potential explanation for the paucity of rotifers on epizoon of U. tumidus. If you find studies in support, please cite them.
Response: So far, works on the epizoon on freshwater bivalves are so rare that if anyone try to deal with this subject at all, there will be first of all - we.
Remark: Also, if fishes are feeding on epizoon of U. tumidus, that might be an opportunity for the attachment of glochidia to potential host fishes. You could discuss this and cite published studies if they are available on glochidia release in Unio tumidus (e.g., Aldridge & McIvor 2003). Aldridge, D.C. & McIvor, A.N. (2003). Gill evacuation and release of glochidia by Unio pictorum and Unio tumidus (bivalvia: unionidae) under thermal and hypoxic stress. Journal of Molluscan Studies 69: 55-59.
Response: Certainly, this is a great idea for a very interesting research. However, it does not seem to have anything to do with the subject of this work, especially since there were no fish in the experiment.
Round 2
Reviewer 1 Report
Accept
Author Response
The manuscript was linguistically checked several times and the other reviewers had no objections. We don't know what else we can do.
Reviewer 3 Report
Please re-visit my original comments and address all the suggested revisions. You have addressed most of the editorial comments and some of the discussion points but you have failed to address important discussion points and points I raised in the Materials and methods. Specifically, in the discussion, you have not addressed points about the potential of bivalves feeding on rotifers, etc. and in the Materials and Methods, you have not explained how microcrustacean and rotifer taxa were identified to species. What resources (cite published references) were used to identify taxa to species? (e.g. Field Guides, taxonomic keys, etc.) You also need to set the alpha significance level in your statistical methods. You have split Figure 1 to make it slightly easier to read, but to me, I would rather see a less jumbled figure using the layout I took the time to suggest to make things easier for your readers to understand. There is just too many graphs overlayed to easily follow. As for Figure 2, I am still a bit unclear on why you chose to present it this way and this needs to be better explained. Also, if you insist on presenting Figure 2 in the way you have, it would be easier to read if you split the figure into two so that Chydorus sphaericus appears separate to the other taxa and you adjust the scale for the other taxa so that density on the y-axis is 0 to 5 so that the trends are more easily discerned. The way the figure looks right now, the other taxa are so compressed given the scale, the graph lines for them are barely legible. Alternatively, you could cut the x-axis down to 30 days and just state in the figure legend that C. sphaericus reached ca. 60 in. 100 cm-2 at Day 40 and was still rising at the end of the experiment.
In the Abstract, spell out species names in full for Dreissena polymorpha and Unio tumidus on first mention, then abbreviate as D. polymorpha and U. tumidus thereafter.
As the manuscript stands, I cannot recommend it for publication until the points above are fully rectified.
Author Response
Remarks and our response to them:
Please re-visit my original comments and address all the suggested revisions. You have addressed most of the editorial comments and some of the discussion points but you have failed to address important discussion points and points I raised in the Materials and methods.
Response: Many thanks for your suggestions. I tried to address all your comments, sometimes presenting my opinion, sometimes changing a fragment of the text in the manuscript. I thought that, as an author, I had the right to stick to my opinion in some cases. However, I will try to adapt to your requirements (especially taking into account the last sentence in your opinion) to the extent that I consider acceptable. If I previously lost one or two (?) of your remarks, please forgive me. I did not do it on purpose.
. Specifically, in the discussion, you have not addressed points about the potential of bivalves feeding on rotifers, etc.
Response: Thanks. We have added this paragraph: “The decrease of the rotifers in the experiment might be related to the diet of both mussels. Makhutova et al. [21] showed that U. tumidus and Dreissena species obtained food of different qualities. In their experiments Dreissena consumed plankton species, i.e., more-valuable food, while U. tumidus fed on detritus and phytobenthic species. Although the diet of both species consisted mainly of algae and detritus enriched with bacteria [22, 23], zooplankton can also be a potential food source. Rotifers had suitable size range, but they would not have been able to reach such high numbers if they had been an essential part of their hosts' diet.”
…and in the Materials and Methods, you have not explained how microcrustacean and rotifer taxa were identified to species. What resources (cite published references) were used to identify taxa to species? (e.g. Field Guides, taxonomic keys, etc.)
Response: Ok thanks. We added two sentences “Rotifers were identified to the species using appropriate identification keys [10 – 17] and Rotifer World Catalog [18]. Identification of crustacean species was based on Janetzky et al. [19], Flößner [20], Błędzki and Rybak [21] etc.”
You also need to set the alpha significance level in your statistical methods.
Response: Done.
You have split Figure 1 to make it slightly easier to read, but to me, I would rather see a less jumbled figure using the layout I took the time to suggest to make things easier for your readers to understand. There is just too many graphs overlayed to easily follow.
Response: I understand your objections. However, I spent a great deal of time trying different ways to illustrate the succession of several of the most important species so that they could be compared with each other according to host species and trophic level, and the last version seems the most appropriate one. After the species were separated, it would be easier to trace the changes in each of them, but more difficult to compare their dynamics. This problem always arises when there are too many data.
As for Figure 2, I am still a bit unclear on why you chose to present it this way and this needs to be better explained. Also, if you insist on presenting Figure 2 in the way you have, it would be easier to read if you split the figure into two so that Chydorus sphaericus appears separate to the other taxa and you adjust the scale for the other taxa so that density on the y-axis is 0 to 5 so that the trends are more easily discerned. The way the figure looks right now, the other taxa are so compressed given the scale, the graph lines for them are barely legible. Alternatively, you could cut the x-axis down to 30 days and just state in the figure legend that C. sphaericus reached ca. 60 in. 100 cm-2 at Day 40 and was still rising at the end of the experiment.
Response: Yes, you are right. We’ve changed it and it is actually much better.
In the Abstract, spell out species names in full for Dreissena polymorpha and Unio tumidus on first mention, then abbreviate as D. polymorpha and U. tumidus thereafter.
Response: OK. Done
As the manuscript stands, I cannot recommend it for publication until the points above are fully rectified.
Response: Really appreciate your suggestions. Please see our responses above and our revised manuscript attached.
Jolanta
Round 3
Reviewer 3 Report
I am satisfied with the changes and recommend the revised version for publication.